# Argmax Flows and Multinomial Diffusion: Learning Categorical Distributions

**Emiel Hoogeboom**[1*]**, Didrik Nielsen**[2*]**, Priyank Jaini**[1]**, Patrick Forré**[3]**, Max Welling**[1]

UvA-Bosch Delta Lab, University of Amsterdam[1],
Technical University of Denmark[2], University of Amsterdam[3]
didni@dtu.dk, e.hoogeboom@uva.nl, p.jaini@uva.nl,
p.d.forre@uva.nl, m.welling@uva.nl

## Abstract

Generative flows and diffusion models have been predominantly trained on ordinal data, for example natural images. This paper introduces two extensions of flows and diffusion for *categorical* data such as language or image segmentation: *Argmax Flows* and *Multinomial Diffusion*. Argmax Flows are defined by a composition of a continuous distribution (such as a normalizing flow), and an argmax function. To optimize this model, we learn a probabilistic inverse for the argmax that lifts the categorical data to a continuous space. Multinomial Diffusion gradually adds categorical noise in a diffusion process, for which the generative denoising process is learned. We demonstrate that our method outperforms existing dequantization approaches on text modelling and modelling on image segmentation maps in log-likelihood.

## 1 Introduction

Many sources of high-dimensional data are categorical, for example language and image segmentation. Although natural images have been studied to a large extent with generative flows and diffusion models, categorical data has not had the same extensive treatment. Currently they are primarily modelled by autoregressive models, which are expensive to sample from (Cooijmans et al., 2017; Dai et al., 2019).

Normalizing flows are attractive because they can be designed to be fast both in the evaluation and sampling direction. Typically, normalizing flows model continuous distributions. As a result, directly optimizing a flow on discrete data may lead to arbitrarily high likelihoods. In literature this problem is resolved for ordinal data by adding noise in a unit interval around the dis-

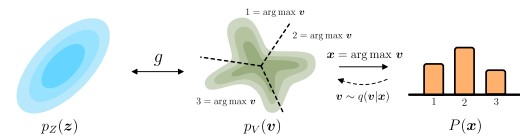

(a) Argmax Flow: Composition of a flow $p(\boldsymbol{v})$ and argmax transformation which gives the model $P(\boldsymbol{x})$. The flow maps from a base distribution $p(\boldsymbol{z})$ using a bijection $g$.

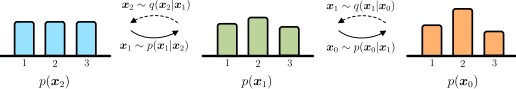

(b) Multinomial Diffusion: Each step $p(\boldsymbol{x}_{t-1}|\boldsymbol{x}_t)$ denoises the signal starting from a uniform categorical base distribution which gives the model $p(\boldsymbol{x}_0)$.

Figure 1: Overview of generative models.

crete value (Uria et al., 2013; Theis et al., 2016; Ho et al., 2019). However, because these methods have been designed for ordinal data, they do not work well on categorical data.

Other attractive generative models are diffusion models (Sohl-Dickstein et al., 2015), which are fast to train due to an objective that decomposes over time steps (Ho et al., 2020). Diffusion models

---

[*]Equal contribution.

35th Conference on Neural Information Processing Systems (NeurIPS 2021).

Table 1: Surjective flow layers for applying continuous flow models to discrete data. The layers are deterministic in the generative direction, but stochastic in the inference direction. Rounding corresponds to the commonly-used dequantization for ordinal data.

| Layer | Generation | Inference | Applications |
|---|---|---|---|
| Rounding | $\boldsymbol{x} = \lfloor \boldsymbol{v} \rfloor$ | $\boldsymbol{v} \sim q(\boldsymbol{v}\vert\boldsymbol{x})$ with support $\mathcal{S}(\boldsymbol{x}) = \{\boldsymbol{v}\vert\boldsymbol{x} = \lfloor \boldsymbol{v} \rfloor\}$ | Ordinal Data e.g. images, audio |
| Argmax | $\boldsymbol{x} = \arg\max \boldsymbol{v}$ | $\boldsymbol{v} \sim q(\boldsymbol{v}\vert\boldsymbol{x})$ with support $\mathcal{S}(\boldsymbol{x}) = \{\boldsymbol{v}\vert\boldsymbol{x} = \arg\max \boldsymbol{v}\}$ | Categorical Data e.g. text, segmentation |

typically have a fixed diffusion process that gradually adds noise. This process is complemented by a learnable generative process that denoises the signal. Song et al. (2020); Nichol and Dhariwal (2021) have shown that diffusion models can also be designed for fast sampling. Thus far, diffusion models have been primarily trained to learn ordinal data distributions, such as natural images.

Therefore, in this paper we introduce extensions of flows and diffusion models for categorical variables (depicted in Figure 1): *i)* Argmax Flows bridge the gap between categorical data and continuous normalizing flows using an argmax transformation and a corresponding family of probabilistic inverses for the argmax. In addition *ii)* we introduce Multinomial Diffusion, which is a diffusion model directly defined on categorical variables. Opposed to normalizing flows, defining diffusion for discrete variables directly does not require gradient approximations, because the diffusion trajectory is fixed. As a result of our work, generative normalizing flows and diffusion models can directly learn categorical data.

## 2 Background

**Normalizing Flows**    Given $\mathcal{V} = \mathbb{R}^d$ and $\mathcal{Z} = \mathbb{R}^d$ with densities $p_V$ and $p_Z$ respectively, normalizing flows (Rezende and Mohamed, 2015) learn a bijective and differentiable transformation $g : \mathcal{Z} \to \mathcal{V}$ such that the change-of-variables formula gives the density at any point $\boldsymbol{v} \in \mathcal{V}$:

$$p_V(\boldsymbol{v}) = p_Z(\boldsymbol{z}) \cdot \left|\det \frac{\mathrm{d}\boldsymbol{z}}{\mathrm{d}\boldsymbol{v}}\right|, \qquad \boldsymbol{v} = g(\boldsymbol{z}), \tag{1}$$

where $p_Z$ can be any density (usually chosen as a standard Gaussian). Thus, normalizing flows provide a powerful framework to learn *exact* density functions. However, Equation (1) is restricted to continuous densities.

To learn densities on ordinal discrete data (such as natural images), typically dequantization noise is added (Uria et al., 2013; Theis et al., 2016; Ho et al., 2019). Nielsen et al. (2020) reinterpreted dequantization as a surjective flow layer $\boldsymbol{v} \mapsto \boldsymbol{x}$ that is deterministic in one direction ($\boldsymbol{x} = \mathrm{round}(\boldsymbol{v})$) and stochastic in the other ($\boldsymbol{v} = \boldsymbol{x} + \boldsymbol{u}$ where $\boldsymbol{u} \sim q(\boldsymbol{u}\vert\boldsymbol{x})$). Using this interpretation, dequantization can be seen as a probabilistic right-inverse for the rounding operation in the latent variable model given by:

$$P(\boldsymbol{x}) = \int P(\boldsymbol{x}\vert\boldsymbol{v})p(\boldsymbol{v})\,\mathrm{d}\boldsymbol{v}, \quad P(\boldsymbol{x}\vert\boldsymbol{v}) = \delta(\boldsymbol{x} = \mathrm{round}(\boldsymbol{v})),$$

where round is applied elementwise. In this case, the density model $p(\boldsymbol{v})$ is modeled using a normalizing flow. Learning proceeds by introducing the variational distribution $q(\boldsymbol{v}\vert\boldsymbol{x})$ that models the probabilistic right-inverse for the rounding surjection and optimizing the evidence lower bound (ELBO):

$$\log P(\boldsymbol{x}) \geq \mathbb{E}_{\boldsymbol{v}\sim q(\boldsymbol{v}\vert\boldsymbol{x})}\left[\log P(\boldsymbol{x}\vert\boldsymbol{v}) + \log p(\boldsymbol{v}) - \log q(\boldsymbol{v}\vert\boldsymbol{x})\right] = \mathbb{E}_{\boldsymbol{v}\sim q(\boldsymbol{v}\vert\boldsymbol{x})}\left[\log p(\boldsymbol{v}) - \log q(\boldsymbol{v}\vert\boldsymbol{x})\right]. \tag{2}$$

The last equality holds under the constraint that the support of $q(\boldsymbol{v}\vert\boldsymbol{x})$ is enforced to be only over the region $\mathcal{S} = \{\boldsymbol{v} \in \mathbb{R}^d : \boldsymbol{x} = \mathrm{round}(\boldsymbol{v})\}$ which ensures that $P(\boldsymbol{x}\vert\boldsymbol{v}) = 1$.

**Diffusion Models**    Given data $\boldsymbol{x}_0$, a diffusion model (Sohl-Dickstein et al., 2015) consists of predefined variational distributions $q(\boldsymbol{x}_t\vert\boldsymbol{x}_{t-1})$ that gradually add noise over time steps $t \in \{1, \ldots, T\}$. The diffusion trajectory is defined such that $q(\boldsymbol{x}_t\vert\boldsymbol{x}_{t-1})$ adds a small amount of noise around $\boldsymbol{x}_{t-1}$. This way, information is gradually destroyed such that at the final time step, $\boldsymbol{x}_T$ carries almost no information about $\boldsymbol{x}_0$. Their generative counterparts consists of learnable distributions $p(\boldsymbol{x}_{t-1}\vert\boldsymbol{x}_t)$ that learn to denoise the data. When the diffusion process adds sufficiently small amounts of noise, it

| **Algorithm 1** Sampling from Argmax Flows | **Algorithm 2** Optimizing Argmax Flows |
|---|---|
| **Input:** $p(\boldsymbol{v})$ 
 **Output:** Sample $\boldsymbol{x}$ 
 Sample $\boldsymbol{v} \sim p(\boldsymbol{v})$ 
 Compute $\boldsymbol{x} = \arg\max \boldsymbol{v}$ | **Input:** $\boldsymbol{x}, p(\boldsymbol{v}), q(\boldsymbol{v}\|\boldsymbol{x})$ 
 **Output:** ELBO $\mathcal{L}$ 
 Sample $\boldsymbol{v} \sim q(\boldsymbol{v}\|\boldsymbol{x})$ 
 Compute $\mathcal{L} = \log p(\boldsymbol{v}) - \log q(\boldsymbol{v}\|\boldsymbol{x})$ |

suffices to define the denoising trajectory using distributions that are factorized (without correlation) over the dimension axis. The distribution $p(\boldsymbol{x}_T)$ is chosen to be similar to the distribution that the diffusion trajectory approaches. Diffusion models can be optimized using variational inference:

$$\log P(\boldsymbol{x}_0) \geq \mathbb{E}_{x_1,\ldots x_T \sim q}\Big[ \log p(\boldsymbol{x}_T) + \sum_{t=1}^{T} \log \frac{p(\boldsymbol{x}_{t-1}|\boldsymbol{x}_t)}{q(\boldsymbol{x}_t|\boldsymbol{x}_{t-1})}\Big].$$

An important insight in diffusion is that by conditioning on $\boldsymbol{x}_0$, the posterior probability $q(\boldsymbol{x}_{t-1}|\boldsymbol{x}_t, \boldsymbol{x}_0) = q(\boldsymbol{x}_t|\boldsymbol{x}_{t-1})q(\boldsymbol{x}_{t-1}|\boldsymbol{x}_0)/q(\boldsymbol{x}_t|\boldsymbol{x}_0)$ is tractable and straightforward to compute, permitting a reformulation in terms of KL divergences that has lower variance (Sohl-Dickstein et al., 2015). Note that $\text{KL}\big(q(\boldsymbol{x}_T|\boldsymbol{x}_0)|p(\boldsymbol{x}_T)\big) \approx 0$ if the diffusion trajectory $q$ is defined well:

$$\log P(\boldsymbol{x}_0) \geq \mathbb{E}_q\Big[ \log p(\boldsymbol{x}_0|\boldsymbol{x}_1) - \text{KL}\big(q(\boldsymbol{x}_T|\boldsymbol{x}_0)|p(\boldsymbol{x}_T)\big) - \sum_{t=2}^{T} \text{KL}\big(q(\boldsymbol{x}_{t-1}|\boldsymbol{x}_t, \boldsymbol{x}_0)|p(\boldsymbol{x}_{t-1}|\boldsymbol{x}_t)\big)\Big] \quad (3)$$

## 3 Argmax Flows

Argmax flows define discrete distributions using 1) a density model $p(\boldsymbol{v})$, such as a normalizing flow, and 2) an argmax layer that maps the continuous $\boldsymbol{v} \in \mathbb{R}^{D \times K}$ to a discrete $\boldsymbol{x} \in \{1, 2, ..., K\}^D$ using

$$\boldsymbol{x} = \arg\max \boldsymbol{v} \quad \text{where} \quad x_d = \arg\max_k v_{dk}. \quad (4)$$

This is a natural choice to model categorical variables, because it divides the entire continuous space of $\boldsymbol{v}$ into symmetric partitions corresponding to categories in $\boldsymbol{x}$. To sample from an argmax flow sample $\boldsymbol{v} \sim p(\boldsymbol{v})$ and compute $\boldsymbol{x} = \arg\max \boldsymbol{v}$ (Algorithm 1). To generate reasonable samples, it is up to the density model $p(\boldsymbol{v})$ to capture any complicated dependencies between the different dimensions. While sampling from an argmax flow is straightforward, the main difficulty lies in *optimizing* this generative model. To compute the likelihood of a datapoint $\boldsymbol{x}$, we have to compute

$$P(\boldsymbol{x}) = \int P(\boldsymbol{x}|\boldsymbol{v})p(\boldsymbol{v})d\boldsymbol{v}, \quad P(\boldsymbol{x}|\boldsymbol{v}) = \delta\big(\boldsymbol{x} = \arg\max(\boldsymbol{v})\big), \quad (5)$$

which is intractable. Consequently, we resort to variational inference and specify a variational distribution $q(\boldsymbol{v}|\boldsymbol{x})$. We note that naïvely choosing any variational distribution may lead to samples $\boldsymbol{v} \sim q(\boldsymbol{v}|\boldsymbol{x})$ where $\delta(\boldsymbol{x} = \arg\max \boldsymbol{v}) = 0$, which yields an ELBO of negative infinity. To avoid this, we need a variational distribution $q(\boldsymbol{v}|\boldsymbol{x})$ that satisfies what we term the *argmax constraint*:

$$\boldsymbol{x} = \arg\max \boldsymbol{v} \quad \text{for all} \quad \boldsymbol{v} \sim q(\boldsymbol{v}|\boldsymbol{x}).$$

That is, the variational distribution $q(\boldsymbol{v}|\boldsymbol{x})$ should have support limited to $\mathcal{S}(\boldsymbol{x}) = \{\boldsymbol{v} \in \mathbb{R}^{D \times K} : \boldsymbol{x} = \arg\max \boldsymbol{v}\}$. Recall that under this condition, the ELBO simplifies to $\mathbb{E}_{\boldsymbol{v} \sim q(\boldsymbol{v}|\boldsymbol{x})}[\log p(\boldsymbol{v}) - \log q(\boldsymbol{v}|\boldsymbol{x})]$, as shown in Algorithm 2. For an illustration of the method see Figure 1a.

### 3.1 Probabilistic Inverse

The argmax layer may be viewed as a surjective flow layer (Nielsen et al., 2020). With this view, the variational distribution $q(\boldsymbol{v}|\boldsymbol{x})$ specifies a distribution over the possible right-inverses of the argmax function, also known as a *stochastic inverse* or *probabilistic inverse*. Recall that the commonly-used dequantization layer for ordinal data corresponds to the probabilistic inverse of a rounding operation. As summarized in Table 1, this layer may thus be viewed as analogous to the argmax layer, where the round is for ordinal data while the argmax is for categorical data.

We are free to specify any variational distribution $q(\boldsymbol{v}|\boldsymbol{x})$ that satisfies the argmax constraint. In the next paragraphs we outline three possible approaches. Since operations are performed independently across dimensions, we omit the dimension axis and let $\boldsymbol{v} \in \mathbb{R}^K$ and $x \in \{1, \ldots, K\}$.

| **Algorithm 3** Thresholding-based $q(\boldsymbol{v}|\boldsymbol{x})$ | **Algorithm 4** Gumbel-based $q(\boldsymbol{v}|\boldsymbol{x})$ |
|---|---|
| **Input:** $\boldsymbol{x}, q(\boldsymbol{u}|\boldsymbol{x})$ 
 **Output:** $\boldsymbol{v}, \log q(\boldsymbol{v}|\boldsymbol{x})$ 
 $\boldsymbol{u} \sim q(\boldsymbol{u}|\boldsymbol{x})$ 
 $\boldsymbol{v}_{\boldsymbol{x}} = \boldsymbol{u}_{\boldsymbol{x}}$ 
 $\boldsymbol{v}_{-\boldsymbol{x}} = \mathrm{threshold}(\boldsymbol{u}_{-\boldsymbol{x}}, \boldsymbol{x})$ 
 $\log q(\boldsymbol{v}|\boldsymbol{x}) = \log q(\boldsymbol{u}|\boldsymbol{x}) - \log|\det \mathrm{d}\boldsymbol{v}/\mathrm{d}\boldsymbol{u}|$ | **Input:** $\boldsymbol{x}, \boldsymbol{\phi}$ 
 **Output:** $\boldsymbol{v}, \log q(\boldsymbol{v}|\boldsymbol{x})$ 
 $\phi_{\max} = \log \sum_i \exp \phi_i$ 
 $\boldsymbol{v}_{\boldsymbol{x}} \sim \mathrm{Gumbel}(\phi_{\max})$ 
 $\boldsymbol{v}_{-\boldsymbol{x}} \sim \mathrm{TruncGumbel}(\boldsymbol{\phi}_{-\boldsymbol{x}}, \boldsymbol{v}_{\boldsymbol{x}})$ 
 $\log q(\boldsymbol{v}|\boldsymbol{x}) = \log \mathrm{Gumbel}(\boldsymbol{v}_{\boldsymbol{x}}|\phi_{\max})$ 
 $\qquad\qquad + \log \mathrm{TruncGumbel}(\boldsymbol{v}_{-\boldsymbol{x}}|\boldsymbol{\phi}_{-\boldsymbol{x}}, \boldsymbol{v}_{\boldsymbol{x}})$ |

**Thresholding (Alg. 3).** A straightforward method to construct a distribution $q(\boldsymbol{v}|x)$ satisfying the argmax constraint is to use thresholding. That is, we first sample an unbounded variable $\boldsymbol{u} \in \mathbb{R}^K$ from $q(\boldsymbol{u}|x)$, which can be for example a conditional Gaussian or normalizing flow. Next, we map $\boldsymbol{u}$ to $\boldsymbol{v}$ such that element $x$ is the largest:

$$v_x = u_x \quad \text{and} \quad \boldsymbol{v}_{-x} = \mathrm{threshold}_T(\boldsymbol{u}_{-x}) \tag{6}$$

where the thresholding is applied elementwise with threshold value $T = v_x$. This ensures that element $v_x$ is the largest, and consequently that $q(\boldsymbol{v}|x)$ satisfies the argmax constraint. Note that we require the threshold function to be bijective, $\mathrm{threshold}_T : \mathbb{R} \to (-\infty, T)$, so that we can use the change-of-variables formula to compute $\log q(\boldsymbol{v}|x)$. In our implementation, thresholding is implemented using a softplus such that all values are mapped below a limit $T$:

$$v = \mathrm{threshold}_T(u) = T - \mathrm{softplus}(T - u), \tag{7}$$

where $\mathrm{softplus}(z) = \log(1 + e^z)$ and for which it is guaranteed that $v \in (-\infty, T)$.

**Gumbel (Alg. 4).** An alternative approach is to let $q(\boldsymbol{v}|x) = \mathrm{Gumbel}(\boldsymbol{v}|\boldsymbol{\phi})$ *restricted* to $\arg\max \boldsymbol{v} = x$, where the location parameters $\boldsymbol{\phi} \leftarrow \mathrm{NN}(x)$ are predicted using a neural network NN. The Gumbel distribution has favourable properties: The $\arg\max$ and $\max$ are independent and the $\max$ is also distributed as a Gumbel:

$$\max_i v_i \sim \mathrm{Gumbel}(\phi_{\max}), \tag{8}$$

where $\phi_{\max} = \log \sum_i \exp \phi_i$. For a more extensive introduction see (Maddison et al., 2014; Kool et al., 2019). To sample $\boldsymbol{v} \sim q(\boldsymbol{v}|x)$, we thus first sample the maximum $v_x$ according to Eq. 8. Next, given the sample $v_x$, the remaining values can be sampled using *truncated* Gumbel distributions:

$$v_i \sim \mathrm{TruncGumbel}(\phi_i; T) \text{ where } i \neq x \tag{9}$$

where the truncation value $T$ is given by $v_x$ which ensures that the argmax constraint $v_x > v_i$ for $i \neq x$ is satisfied. Recall that to optimize Eq. 2, $\log q(\boldsymbol{v}|x)$ is also required, which can be computed using the closed-form expressions for the log density functions (see Table 5). Another property of Gumbel distributions is that

$$P(\arg\max \boldsymbol{v} = i) = \exp \phi_i / \sum_i \exp \phi_i, \tag{10}$$

which we use to initialize the location parameters $\boldsymbol{\phi}$ to match the empirical distribution of the first minibatch of the data.

**Gumbel Thresholding.** This method unifies the methods from the previous two sections: Gumbel distributions and thresholding. The key insight is that the Gumbel sampling procedures as defined above can be seen as a reparametrization of a uniform noise distribution $\mathcal{U}(0, 1)^K$ which is put through the inverse CDF of the Gumbel distributions (see Table 5). From the perspective of change-of-variables, the log likelihood denotes the log volume change of this transformation. To increase expressivity the uniform distribution can be replaced by a normalizing flow $q(\boldsymbol{u}|x)$ that has support on the interval $(0, 1)^K$, which can be enforced using a sigmoid transformation. This section shows that a large collection of thresholding functions can be found by studying (truncated) inverse CDFs. In practice we find that performance is reasonably similar as long as the underlying noise $\boldsymbol{u}$ is learned.

**Behavior of the Variational Posterior**   Although several methods to learn $q$ have been proposed, it is unclear what expressivity is required. In the following, the interactions between $q(\boldsymbol{v}|\boldsymbol{x})$ and the density model $p(\boldsymbol{v})$ are discussed. Recall that the variational bound that is optimized under expectation of a data distribution $\mathcal{D}$ can be seen as minimizing the KL distance between the aggregated posterior $q(\boldsymbol{v}) = \mathbb{E}_{\boldsymbol{x} \sim \mathcal{D}} q(\boldsymbol{v}|\boldsymbol{x})$ and the density model $p(\boldsymbol{v})$, so $\mathrm{KL}(q(\boldsymbol{v})|p(\boldsymbol{v}))$. There are two distinct reasons which can cause this distance to be large: Firstly, the density model $p(\boldsymbol{v})$ may not have the right probability mass in each argmax region. These desired probabilities solely depend on the data distribution $\mathcal{D}$. Secondly, the variational posterior $q(\boldsymbol{v}|\boldsymbol{x})$ may not have the correct shape compared to $p(\boldsymbol{v})$, *within* an argmax region. At initialization, the thresholding within $q$ can create low density regions at argmax boundaries.

In theory, if $p(\boldsymbol{v})$ is a universal density approximator, then the model can be fitted for any well-behaved $q(\boldsymbol{v}|\boldsymbol{x})$. Then $p(\boldsymbol{v})$ can even fit the low density regions in the boundaries. This argument is trivial, as one can simply set $p(\boldsymbol{v})$ to $q(\boldsymbol{v}) = \mathbb{E}_{\boldsymbol{x} \sim \mathcal{D}} q(\boldsymbol{v}|\boldsymbol{x})$. In practice, over training steps we find that $q$ does smooth out these boundary artifacts, and counteracts the thresholding so that the aggregated posterior becomes smoother.

### 3.2   Cartesian Products of Argmax Flows

In the current description, Argmax Flows require the same number of dimensions in $\boldsymbol{v}$ as there are classes in $\boldsymbol{x}$. To alleviate this constraint we introduce Cartesian products of Argmax Flows. To illustrate our method, consider a 256 class problem. One class can be represented using a single number in $\{1, \dots, 256\}$, but also using two hexadecimal numbers $\{1, \dots, 16\}^2$ or alternatively using eight binary numbers. Specifically, any base $K$ variable $\boldsymbol{x}^{(K)} \in \{1, \dots, K\}^D$ can be converted to a base $M$ variable $\boldsymbol{x}^{(M)} \in \{1, \dots, M\}^{d_m \times D}$ where $d_m = \lceil \log_M K \rceil$. Then the variable $\boldsymbol{x}^{(M)}$ with dimensionality $M \cdot d_m \cdot D$ represents the variable $\boldsymbol{x}^{(K)}$ with dimensionality $K \cdot D$, trading off symmetry for dimensionality. Even though this may lead to some unused additional classes, the ELBO objective in Equation 2 can still be optimized using an $M$-categorical Argmax Flow. Finally, note that Cartesian products of binary spaces are a special case where the variable can be encoded symmetrically into a single dimension to the positive and negative part using binary dequantization (Winkler et al., 2019). In this case, by trading-off symmetry the dimensionality increases only proportional to $\log_2 K$ .

## 4   Multinomial Diffusion

In this section we introduce an alternative likelihood-based model for categorical data: Multinomial Diffusion. In contrast with previous sections, $\boldsymbol{x}_t$ will be represented in one-hot encoded format $\boldsymbol{x}_t \in \{0, 1\}^K$. Specifically, for category $k$, $x_k = 1$ and $x_j = 0$ for $j \neq k$. Note that again the dimension axis is omitted for clarity as all distributions are independent over the dimension axis. We define the multinomial diffusion process using a categorical distribution that has a $\beta_t$ chance of resampling a category uniformly:

$$q(\boldsymbol{x}_t|\boldsymbol{x}_{t-1}) = \mathcal{C}(\boldsymbol{x}_t|(1 - \beta_t)\boldsymbol{x}_{t-1} + \beta_t/K), \tag{11}$$

where $\mathcal{C}$ denotes a categorical distribution with probability parameters after $|$. Further addition (and subtraction) between scalars and vectors is done elementwise. This convention kept throughout this section. Since these distributions form a Markov chain, we can express the probability of any $\boldsymbol{x}_t$ given $\boldsymbol{x}_0$ as:

$$q(\boldsymbol{x}_t|\boldsymbol{x}_0) = \mathcal{C}(\boldsymbol{x}_t|\bar{\alpha}_t \boldsymbol{x}_0 + (1 - \bar{\alpha}_t)/K) \tag{12}$$

where $\alpha_t = 1 - \beta_t$ and $\bar{\alpha}_t = \prod_{\tau=1}^t \alpha_\tau$. Intuitively, for each next timestep, a little amount of uniform noise $\beta_t$ over the $K$ classes is introduced, and with a large probability $(1 - \beta_t)$ the previous value $\boldsymbol{x}_{t-1}$ is sampled. Using Equation 11 and 12 the categorical posterior $q(\boldsymbol{x}_{t-1}|\boldsymbol{x}_t, \boldsymbol{x}_0)$ can be computed in closed-form:

$$q(\boldsymbol{x}_{t-1}|\boldsymbol{x}_t, \boldsymbol{x}_0) = \mathcal{C}(\boldsymbol{x}_{t-1}|\boldsymbol{\theta}_{\mathrm{post}}(\boldsymbol{x}_t, \boldsymbol{x}_0)), \quad \text{where} \quad \boldsymbol{\theta}_{\mathrm{post}}(\boldsymbol{x}_t, \boldsymbol{x}_0) = \tilde{\boldsymbol{\theta}}/\sum_{k=1}^K \tilde{\theta}_k \tag{13}$$

$$\text{and} \quad \tilde{\boldsymbol{\theta}} = [\alpha_t \boldsymbol{x}_t + (1 - \alpha_t)/K] \odot [\bar{\alpha}_{t-1} \boldsymbol{x}_0 + (1 - \bar{\alpha}_{t-1})/K].$$

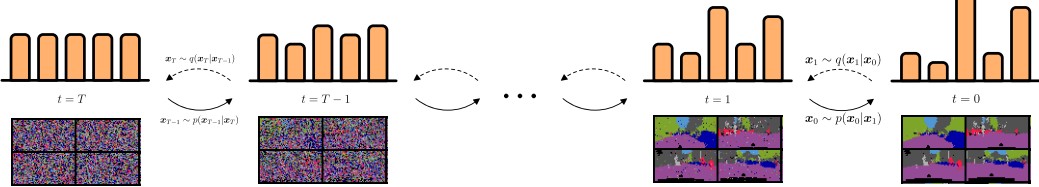

Figure 2: Overview of multinomial diffusion. A generative model $p(\boldsymbol{x}_{t-1}|\boldsymbol{x}_t)$ learns to gradually denoise a signal from left to right. An inference diffusion process $q(\boldsymbol{x}_t|\boldsymbol{x}_{t-1})$ gradually adds noise form right to left.

One of the innovations in Ho et al. (2020) was the insight to not predict the parameters for the generative trajectory directly, but rather to predict the noise using the posterior equation for $q$. Although predicting the noise is difficult for discrete data, we predict a probability vector for $\hat{\boldsymbol{x}}_0$ from $\boldsymbol{x}_t$ and subsequently parametrize $p(\boldsymbol{x}_{t-1}|\boldsymbol{x}_t)$ using the probability vector from $q(\boldsymbol{x}_{t-1}|\boldsymbol{x}_t, \hat{\boldsymbol{x}}_0)$, where $\boldsymbol{x}_0$ is approximated using a neural network $\hat{\boldsymbol{x}}_0 = \mu(\boldsymbol{x}_t, t)$. Equation 13 will produce valid probability vectors that are non-negative and sums to one under the condition that the prediction $\hat{\boldsymbol{x}}_0$ is non-negative and sums to one, which is ensured with a softmax function in $\mu$. To summarize:

$$p(\boldsymbol{x}_0|\boldsymbol{x}_1) = \mathcal{C}(\boldsymbol{x}_0|\hat{\boldsymbol{x}}_0) \text{ and } p(\boldsymbol{x}_{t-1}|\boldsymbol{x}_t) = \mathcal{C}(\boldsymbol{x}_{t-1}|\boldsymbol{\theta}_{\text{post}}(\boldsymbol{x}_t, \hat{\boldsymbol{x}}_0)) \text{ where } \hat{\boldsymbol{x}}_0 = \mu(\boldsymbol{x}_t, t) \quad (14)$$

The KL terms in Equation 3 can be simply computed by enumerating the probabilities in Equation 13 and 14 and computing the KL divergence for discrete distributions in $L_{t-1}$ with $t \geq 2$:

$$\text{KL}\big(q(\boldsymbol{x}_{t-1}|\boldsymbol{x}_t, \boldsymbol{x}_0)|p(\boldsymbol{x}_{t-1}|\boldsymbol{x}_t)\big) = \text{KL}\big(\mathcal{C}(\boldsymbol{\theta}_{\text{post}}(\boldsymbol{x}_t, \boldsymbol{x}_0))|\mathcal{C}(\boldsymbol{\theta}_{\text{post}}(\boldsymbol{x}_t, \hat{\boldsymbol{x}}_0))\big), \quad (15)$$

which can be computed using $\sum_k \boldsymbol{\theta}_{\text{post}}(\boldsymbol{x}_t, \boldsymbol{x}_0))_k \cdot \log \frac{\boldsymbol{\theta}_{\text{post}}(\boldsymbol{x}_t, \boldsymbol{x}_0))_k}{\boldsymbol{\theta}_{\text{post}}(\boldsymbol{x}_t, \hat{\boldsymbol{x}}_0))_k}$. Furtermore, to compute $\log p(\boldsymbol{x}_0|\boldsymbol{x}_1)$ use that $\boldsymbol{x}_0$ is onehot:

$$\log p(\boldsymbol{x}_0|\boldsymbol{x}_1) = \sum_k \boldsymbol{x}_{0,k} \log \hat{\boldsymbol{x}}_{0,k} \quad (16)$$

## 5  Related Work

Deep generative models broadly fall into the categories autoregressive models ARMs (Germain et al., 2015), Variational Autoencoders (VAEs) (Kingma and Welling, 2014; Rezende et al., 2014), Adversarial Network (GANs) (Goodfellow et al., 2014), Normalizing Flows (Rezende and Mohamed, 2015), Energy-Based Models (EBMs) and Diffusion Models (Sohl-Dickstein et al., 2015).

Normalizing Flows typically learn a continuous distribution and dequantization is required to train these methods on ordinal data such as images. A large body of work is dedicated to building more expressive continuous normalizing flows (Dinh et al., 2017; Germain et al., 2015; Kingma et al., 2016; Papamakarios et al., 2017; Chen et al., 2018; Song et al., 2019; Perugachi-Diaz et al., 2020). To learn ordinal discrete distributions with normalizing flows, adding uniform noise in-between ordinal classes was proposed in (Uria et al., 2013) and later theoretically justified in (Theis et al., 2016). An extension for more powerful dequantization based on variational inference was proposed in (Ho et al., 2019), and connected to autoregressive models in (Nielsen and Winther, 2020). Dequantization for binary variables was proposed in (Winkler et al., 2019). Tran et al. (2019) propose invertible transformations for categorical variables directly. However, these methods can be difficult to train because of gradient bias and results on images have thus far not been demonstrated. In addition flows for ordinal discrete data (integers) have been explored in (Hoogeboom et al., 2019; van den Berg et al., 2020). In other works, VAEs have been adapted to learn a normalizing flow for the latent space (Ziegler and Rush, 2019; Lippe and Gavves, 2020). However, these approaches typically still utilize an argmax heuristic to sample, even though this is not the distribution specified during training.

Diffusion models were first introduced in Sohl-Dickstein et al. (2015), who developed diffusion for Gaussian and Bernoulli distributions. Recently, Denoising Diffusion models Ho et al. (2020) have been shown capable of generating high-dimensional images by architectural improvements and reparametrization of the predictions. Diffusion models are relatively fast to train, but slow to sample

Table 2: Comparison of a coupling and autoregressive generative flows with uniform (Uria et al., 2013) and variational (Ho et al., 2019) dequantization and our proposed Argmax flows.

| Dequantization | Flow type | text8 (bpc) | enwik8 (bits per raw byte) |
|---|---|---|---|
| Uniform dequantization | | 1.90 | 2.14 |
| Variational dequantization | Autoregressive | 1.43 | 1.44 |
| Argmax Flow (ours) | | **1.38** | **1.42** |
| Uniform dequantization | | 2.01 | 2.33 |
| Variational dequantization | Coupling | 2.08 | 2.28 |
| Argmax Flow (ours) | | **1.82** | **1.93** |

from as they require iterations over the many timesteps in the chain. Song et al. (2020); Nichol and Dhariwal (2021) showed that in practice samples can be generated using significantly fewer steps. Nichol and Dhariwal (2021) demonstrated that importance-weighting the objective components greatly improves log-likelihood performance. In Song et al. (2020) a continuous-time extension of denoising diffusion models was proposed. After initial release of this paper we discovered that Song et al. (2020) concurrently also describe a framework for discrete diffusion, but without empirical evaluation.

## 6 Experiments

In our experiments we compare the performance of our methods on language modelling tasks and learning image segmentation maps unconditionally.

### 6.1 Language data

In this section we compare our methods on two language datasets, `text8` and `enwik8`. `text8` contains 27 categories ('a' through 'z' and ' ') and for `enwik8` the bytes are directly modelled which results in 256 categories.

**Model description**   Two versions of generative argmax flows are tested: using an autoregressive (AR) flow and a coupling-based flow for $p(\boldsymbol{v})$. In these experiments the probabilistic inverse is based on the thresholding approach. Specifically, a conditional diagonal Gaussian $q(\boldsymbol{u}|\boldsymbol{x})$ is trained and thresholded which gives the distribution $q(\boldsymbol{v}|\boldsymbol{x})$. The argmax flow is defined on binary Cartesian products. This means that for $K = 27$, a 5-dimensional binary space is used and for $K = 256$ an 8-dimensional binary space. The argmax flow is compared to the current standard of training generative flows directly on discrete data: dequantization. We compare to both uniform and variational dequantization, where noise on a $(0, 1)$ interval is added to the onehot representation of the categorical data. The autoregressive density model is based on the model proposed in (Lippe and Gavves, 2020). The coupling density model consists of 8 flow layers where each layer consists of a $1 \times 1$ convolution and mixture of logistics transformations Ho et al. (2019). In the multinomial text diffusion model, the $\mu$ network is modeled by a 12-layer Transformer. For more extensive details about the experiment setup see Appendix B.

Table 3: Comparison of different methods on `text8` and `enwik8`. Results are reported in negative log-likelihood with units bits per character (bpc) for `text8` and bits per raw byte (bpb) for `enwik8`.

| Model type | Model | text8 (bpc) | enwik8 (bpb) |
|---|---|---|---|
| ARM | 64 Layer Transformer (Al-Rfou et al., 2019) | 1.13 | 1.06 |
| | TransformerXL (Dai et al., 2019) | 1.08 | 0.99 |
| VAE | AF/AF★ (AR) (Ziegler and Rush, 2019) | 1.62 | 1.72 |
| | IAF / SCF★ (Ziegler and Rush, 2019) | 1.88 | 2.03 |
| | CategoricalNF (AR) (Lippe and Gavves, 2020) | 1.45 | - |
| Generative Flow | Argmax Flow, AR (ours) | 1.39 | 1.42 |
| | Argmax Coupling Flow (ours) | 1.82 | 1.93 |
| Diffusion | Multinomial Text Diffusion (ours) | 1.72 | 1.75 |

★ Results obtained by running code from the official repository for the `text8` and `enwik8` datasets.

```
 that the role of tellings not be required also action characters passe
d on constitution ahmad a nobilitis first be closest to the cope and dh
ur and nophosons she criticized itm specifically on august one three mo
vement and a renouncing local party of exte

nt is in this meant the replicat today through the understanding elemen
t thinks the sometimes seven five his final form of contair you are lot
ur and me es to ultimately this work on the future all all machine the
silon words thereis greatly usaged up not t
```

(a) Samples from Multinomial Text Diffusion.

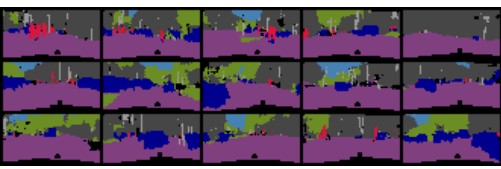

(a) Samples from the Argmax Flow.

```
 heartedness frege thematically infered by the famous existence of a fu
nction f from the laplace definition we can analyze a definition of bin
ary operations with additional size so their functionality cannot be re
viewed here there is no change because its

otal cost of learning objects from language to platonic linguistics exa
mines why animate to indicate wild amphibious substances animal and mar
ine life constituents of animals and bird sciences medieval biology bio
logy and central medicine full discovery re
```

(b) Samples from Argmax AR Flow.

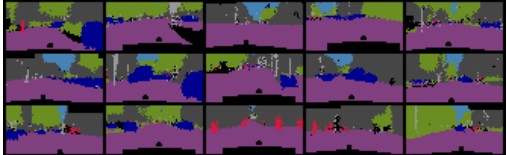

(b) Samples from the Multinomial Diffusion model.

```
ns fergenur d alpha and    le heigu man notabhe leglon lm n two six a gg
opa movement as sympathetic dutch the term bilirubhah acquired the bava
rian cheeh segt thmamouinaire vhvinus lihnos ineoneartis or medical iod
ine the rave wesp published harsy varb hhgh

 danibah or manuccha but calpere that  of the moisture soods and dristi
ng attempt to cause any moderator called lk brown or totpdngs is usuall
y able to nus and hockecrits borel qbisupnias section rybancase untecce
mentation anymore the motion of plays on qr
```

(c) Samples from Argmax Coupling Flow.

Figure 3: Samples from models, `text8`.

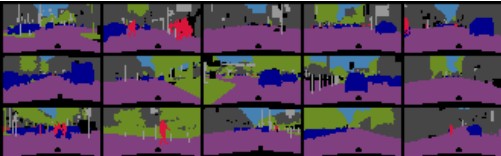

(c) Cityscapes data.

Figure 4: Samples from models, cityscapes.

**Comparison with Generative Flows** Firstly we compare the performance of generative flows directly trained on language data (Table 2). These experiments are using the same underlying normalizing flow: either a coupling-based flow or an autoregressive flow. Note that Argmax Flows consistently outperform both uniform and variational dequantization. This indicates that it is easier for a generative flow to learn the lifted continuous distribution using an argmax flow. An advantage of Argmax flows that may explain this difference is that they lift the variables into the entire Euclidean space, whereas traditional dequantization only introduce probability density on $(0, 1)$ intervals, leaving gaps with no probability density. The performance improvements of Argmax flows are even more pronounced when comparing coupling-based approaches. Also note that coupling flows have worse performance than autoregressive flows, with a difference that is generally smaller for images. This indicates that designing more expressive coupling layers for text is an interesting future research direction.

**Comparison with other generative models** The performance compared to models in literature is presented in Table 3 alongside the performance of our Argmax Flows and Multinomial Diffusion. The latent variable approaches containing autoregressive components are marked using (AR). Although autoregressive flows still have the same disadvantages as ARMs, they provide perspective on where performance deficiencies are coming from. We find that our autoregressive Argmax Flows achieve better performance than the VAE approaches, they outperform AF/AF (Ziegler and Rush, 2019) and CategoricalNF (Lippe and Gavves, 2020).

When comparing non-autoregressive models, Argmax Flows also outperforms the method that lifts the categorical space to a continuous space: IAF / SCF (Ziegler and Rush, 2019). Interestingly, the multinomial text diffusion is a non-autoregressive model that performs even better than the argmax coupling flow, but performs worse than the autoregressive version. For this model it is possible that different diffusion trajectories for $q$ would result in even better performance, because in the current form the denoising model has to be very robust to input noise. These experiments also highlight that there is still a distinct performance gap between standard ARMs and (autoregressive) continuous density model on text, possibly related to the dequantization gap (Nielsen and Winther, 2020). Samples from different models trained on `text8` are depicted in Figure 3. Because of difficulties in reproducing results from Discrete Flows, a comparison and analysis of discrete flows are left out of this section. Instead they are extensively discussed in Appendix C. For additional experiments regarding Cartesian products and sampling time see Appendix D.

**Unsupervised spell-checking** An interesting by-product of the text diffusion model is that it can be used to spell-check text using a single forward pass. To demonstrate this, a sentence taken from the test data is corrupted by changing a few characters. This corrupted sequence is given as $x_1$ to the generative denoising model, which is close to the data at step 0. Then the denoising model predicts $p(x_0|x_1)$ and the most-likely $x_0$ can be suggested. Note that this model only works for character-level corruption, not insertions. An example is depicted in Figure 5. Since the model chooses the most-likely matching word, larger corruptions will at some point lead to word changes.

```
mexico city the aztec stadium estadio azteca home of club america is on
e of the world s largest stadiums with capacity to seat approximately o
ne one zero zero zero zero fans mexico hosted the football world cup in
one nine seven zero and one nine eight six
```

(a) Ground truth sequence from `text8`.

```
mexico citi the aztec stadium estadio azteca home of club amerika is on
e of the world s largest stadioms with capakity to seat approsimately o
ne one zeto zero zero zero fans mexico hosted the footpall wolld cup in
one nine zeven zero and one nyne eigt six
```

(b) Corrupted sentence.

```
mexico city the aztec stadium estadio aztecs home of club america is on
e of the world s largest stadiums with capacity to seat approximately o
ne one zero zero zero zero fans mexico hosted the football world cup in
one nine seven zero and one nine eight six
```

(c) Suggested, prediction by the model.

Figure 5: Spell checking with Multinomial Text Diffusion.

## 6.2 Segmentation maps

For image-type data, we introduce a categorical image dataset: the cityscapes dataset is repurposed for *unconditional* image segmentation learning. In contrast with the standard setting, the distribution over the segmentation targets needs to be learned *without* conditioning on the photograph. To reduce computational cost, we rescale the segmentation maps from cityscapes to $32 \times 64$ images using nearest neighbour interpolation. We utilize the global categories as prediction targets which results in an 8-class problem.

**Model description** The Argmax Flows are defined directly on the $K = 8$ categorical space. The density model $p(v)$ is defined using affine coupling layers parametrized by DenseNets (Huang et al., 2017). For the probabilistic inverse we learn a conditional flow $q(u|x)$ which is also based on the affine coupling structure. Depending on the method, either softplus or Gumbel thresholding is applied to obtain $v$. Recall that for our first Gumbel approach it is equivalent to set $q(u|x)$ to the unit uniform distribution, whereas $q(u|x)$ is learned for Gumbel

Table 4: Performance of different dequantization methods on squares and cityscapes dataset, in bits per pixel, lower is better.

| Cityscapes | ELBO | IWBO |
|---|---|---|
| Round / Unif. (Uria et al., 2013) | 1.010 | 0.930 |
| Round / Var. (Ho et al., 2019) | 0.334 | 0.315 |
| Argmax / Softplus thres. (ours) | **0.303** | **0.290** |
| Argmax / Gumbel dist. (ours) | 0.365 | 0.341 |
| Argmax / Gumbel thres. (ours) | **0.307** | **0.287** |
| Multinomial Diffusion (ours) | 0.305 | |

thresholding. We compare to existing dequantization strategies in literature: uniform (Uria et al., 2013) and variational dequantization (Ho et al., 2019) which are applied on the onehot representation. All models utilize the same underlying flow architectures and thus the number of parameters is roughly the same. The exception are uniform dequantization and the Gumbel distribution, since no additional variational flow distribution is needed. For more extensive details see Appendix B.

**Comparison** The results of this experiment are shown in Table 4 in terms of ELBO and if available the IWBO (importance weighted bound) (Burda et al., 2016) with 1000 samples measured in bits per pixel. Consistent with the language experiments, the traditional dequantization approaches (uniform / variational) are outperformed by Argmax Flows. Interestingly, although argmax flows with softplus thresholding achieves the best ELBO, the argmax flow with Gumbel thresholding approach achieves a better IWBO. The Multinomial Diffusion model performs somewhat worse with 0.37 bpp on test whereas it scored 0.33 bpp on train. Interestingly, this the only model where overfitting was an issue and data augmentation was required, which may explain this portion of the performance difference. For all other models training performance was comparable to test and validation performance. Samples from the different models trained on cityscapes are depicted in Figure 4. Another interesting point is that coupling flows had difficulty producing coherent text samples (Figure 3) but do not suffer from this problem on the cityscapes data which is more image-like. As coupling layers where initially designed for images (Dinh et al., 2015), they may require adjustments to increase their expressiveness on text.

# 7 Social Impact and Conclusion

**Social Impact**   The methods described in this paper can be used to learn categorical distributions. For that reason, they can potentially be used to generate high-dimensional categorical data, such as text or image segmentation maps, faster than iterative approaches. Possibly negative influences are the generation of fake media in the form of text, or very unhelpful automated chat bots for customer service. Our work could positively influence new methods for text generation, or improved segmentation for self-driving cars. In addition, our work may also be used for outlier detection to flag fake content. Also, we believe the method in its current form is still distant from direct applications as the ones mentioned above.

**Conclusion**   In this paper we propose two extensions for Normalizing Flows and Diffusion models to learn categorical data: Argmax Flows and Multinomial Diffusion. Our experiments show that our methods outperform comparable models in terms of negative log-likelihood. In addition, our experiments highlight distinct performance gaps in the field: Between standard ARMs, continuous autoregressive models and non-autoregressive continuous models. This indicates that future work could focus on two sources of decreased performance: 1) when discrete variables are lifted to a continuous space and further 2) when removing autoregressive components.

**Funding Disclosure**
There are no additional sources of funding to disclose, beyond the affiliations of the authors.

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
