# A  Numerically stable Multinomial Diffusion in log space

In this section we explain how Multinomial Diffusion models can be implemented in a numerically safe manner in log-space. Note that in addition to this appendix with pseudo-code, the actual source code will also be released. First we define a few helper functions:

```
def log_add_exp(a, b):
    maximum = max(a, b)
    return maximum + log(exp(a - maximum) + exp(b - maximum))

def log_sum_exp(x):
    maximum = max(x, dim=1, keepdim=True)
    return maximum + log(exp(x - maximum).sum(dim=1))

def index_to_log_onehot(x, num_classes):
    # Assume that onehot axis is inserted at dimension 1
    x_onehot = one_hot(x, num_classes)

    # Compute in log-space, extreme low values are later
    # filtered out by log sum exp calls.
    log_x = log(x_onehot.clamp(min=1e-40))
    return log_x

def log_onehot_to_index(log_x):
    return log_x.argmax(1)

def log_1_min_a(a):
    return log(1 - a.exp() + 1e-40)
```

Then we can initialize the variables we are planning to utilize for the multinomial diffusion model. This is done with float64 variables to limit the precision loss in the `log_1_min_a` computation. Since these are precomputed and later converted to float32, there is no meaningful increase in computation time.

```
alphas = init_alphas()
log_alpha = np.log(alphas)
log_cumprod_alpha = np.cumsum(log_alpha)

log_1_min_alpha = log_1_min_a(log_alpha)
log_1_min_cumprod_alpha = log_1_min_a(log_cumprod_alpha)
```

Then we can define the functions that we utilize to compute the log probabilities of the categorical distributions of the forward process. The functions below compute the probability vectors for $q(\boldsymbol{x}_t|\boldsymbol{x}_{t-1})$, $q(\boldsymbol{x}_t|\boldsymbol{x}_0)$ and $q(\boldsymbol{x}_{t-1}|\boldsymbol{x}_t, \boldsymbol{x}_0)$.

```
def q_pred_one_timestep(log_x_t, t):
    # Computing alpha_t * E[xt] + (1 - alpha_t) 1 / K
    log_probs = log_add_exp(
        log_x_t + log_alpha[t],
        log_1_min_alpha[t] - log(num_classes)
    )
    return log_probs

def q_pred(log_x0, t):
    log_probs = log_add_exp(
        log_x0 + log_cumprod_alpha[t],
        log_1_min_cumprod_alpha[t] - log(num_classes)
    )
    return log_probs

def q_posterior(log_x0, log_x_t, t):
    # Kronecker delta peak for q(x0 | x1, x0).
    if t == 0:
        log_probs_xtmin = log_x0
```

```
    else:
        log_probs_xtmin = q_pred(log_x0, t - 1)

    # Note log_x_t is used not x_tmin, subtle and not straightforward
    # why this is true. Corresponds to Algorithm 1.
    unnormed_logprobs = log_probs_xtmin + q_pred_one_timestep(log_x_t, t)

    log_probs_posterior = unnormed_logprobs - log_sum_exp(unnormed_logprobs)
    return log_probs_posterior
```

Some magic is happening in `q_pred_one_timestep`. Recall that at some point we need to compute $\mathcal{C}(\boldsymbol{x}_t|(1 - \beta_t)\boldsymbol{x}_{t-1} + \beta_t/K)$ for different values of $\boldsymbol{x}_t$, which when treated as a function outputs $(1 - \beta_t) + \beta_t/K$ if $\boldsymbol{x}_t = \boldsymbol{x}_{t-1}$ and $\beta_t/K$ otherwise. This function is symmetric, meaning that $\mathcal{C}(\boldsymbol{x}_t|(1 - \beta_t)\boldsymbol{x}_{t-1} + \beta_t/K) = \mathcal{C}(\boldsymbol{x}_{t-1}|(1 - \beta_t)\boldsymbol{x}_t + \beta_t/K)$. This is why we can switch the conditioning and immediately return the different probability vectors for $\boldsymbol{x}_t$. This also corresponds to Equation 13.

Then using the `q_posterior` function as parametrization we predict the probability vector for $p(\boldsymbol{x}_{t-1}|\boldsymbol{x}_t)$ using a neural network.

```
def p_pred(log_x_t, t):
    x_t = log_onehot_to_index(log_x_t)
    log_x_recon = logsoftmax(neuralnet(x_t, t))
    log_model_pred = q_posterior(log_x_recon, log_x_t, t)
    return log_model_pred
```

And then finally we can compute the loss term $L_t$ using the KL divergence for categorical distributions:

```
def categorical_kl(log_prob_a, log_prob_b):
    kl = (log_prob_a.exp() * (log_prob_a - log_prob_b)).sum(dim=1)
    return kl

def compute_Lt(log_x0, log_x_t, t):
    log_true_prob = q_posterior(log_x0, log_x_t, t)
    log_model_prob = p_pred(log_x_t, t)
    kl = categorical_kl(log_true_prob, log_model_prob)
    loss = sum_except_batch(kl)
    return loss
```

Coincidentally this code even works for $L_0$ because $\boldsymbol{x}_0$ is onehot and then:

$$- \log \mathcal{C}(\boldsymbol{x}_0|\hat{\boldsymbol{x}}_0) - \sum_k \boldsymbol{x}_{0,k} \log \hat{\boldsymbol{x}}_{0,k} = \sum_k \boldsymbol{x}_{0,k} [\underbrace{\log \boldsymbol{x}_{0,k}}_{0 \text{ or } \log 0} - \log \hat{\boldsymbol{x}}_{0,k}] = \text{KL}(\mathcal{C}(\boldsymbol{x}_0)||\mathcal{C}(\hat{\boldsymbol{x}}_0)),$$

where in the last term $\boldsymbol{x}_0$ and $\hat{\boldsymbol{x}}_0$ are probability vectors and $0 \log 0$ is defined to be $0$.

# B Experimental details

This section gives details on experimental setup, architectures and optimization hyperparameters. In addition, the code to reproduce experiments will be released publicly.

**Diffusion settings** For diffusion we use the cosine schedule for $\{\alpha_t\}$ from Nichol and Dhariwal (2021) with the difference that what was previously $\sqrt{\bar{\alpha}_t}$ is now $\bar{\alpha}_t$, so that their factor $\sqrt{\bar{\alpha}_t}$ for the Gaussian mean is equal to our factor $\bar{\alpha}_t$ for categorical parameters. Specifically, our $\bar{\alpha}_t$ are defined using:

$$\bar{\alpha}_t = \frac{f(t)}{f(0)} \quad f(t) = \cos\left(\frac{t/T + s}{1+s} \cdot \frac{\pi}{2}\right), \quad s = 0.008,$$

where $T$ is the total number of diffusion steps. Nichol and Dhariwal (2021) show that instead of sampling $t$ uniformly, variance is reduced when $t$ is importance-sampled with $q(t) \propto \sqrt{\mathbb{E}[L_t^2]}$, which is estimated using training statistics, and we use their approach. The objective can be summarized as:

$$\log P(\boldsymbol{x}_0) \geq \mathbb{E}_{t \sim q(t), \boldsymbol{x}_t \sim q(\boldsymbol{x}_t | \boldsymbol{x}_0)} \left[ -\frac{1}{q(t)} \mathrm{KL}\big(q(\boldsymbol{x}_{t-1} | \boldsymbol{x}_t, \boldsymbol{x}_0) | p(\boldsymbol{x}_{t-1} | \boldsymbol{x}_t)\big) \right]. \tag{17}$$

**Gumbel properties** In Table 5 a useful overview of Gumbel properties are given. These equations can be used to sample and compute the likelihood of the (truncated) Gumbel distributions. For a more extensive treatment see (Maddison et al., 2014; Kool et al., 2019).

Table 5: Summary of Gumbel properties.

| Description | $\log p$ | Sample |
|---|---|---|
| $\mathrm{Gumbel}(g\|\phi)$ | $\phi - g - \exp(\phi - g)$ | $g = -\log(-\log(u)) + \phi$ 
 $u \sim \mathcal{U}(0,1)$ |
| $\max_i \mathrm{Gumbel}(g_i\|\phi)$ | $\log \mathrm{Gumbel}(g_{\max}\|\phi_{\max})$ 
 $\phi_{\max} = \log \sum_i \exp \phi_i$ | $g_{\max} \sim \mathrm{Gumbel}(\phi_{\max})$ 
 $\phi_{\max} = \log \sum_i \exp \phi_i$ |
| $\mathrm{TruncGumbel}(g\|\phi, T)$ | $\phi - g - \exp(\phi - g) + \exp(\phi - T)$ 
 if $g < T$ else $-\infty$ | $g = \phi - \log(\exp(\phi - T) - \log u)$ 
 $u \sim \mathcal{U}(0,1)$ |

## B.1 Language Modelling

For the language modelling experiments we utilize the standard `text8` dataset with sequence length 256 and `enwik8` dataset with sequence length 320. The train/val/test splits are 90000000/5000000/5000000 for both `text8` and `enwik8`, as is standard in literature. The Multinomial Text Diffusion models are trained for 300 epochs, whereas the Argmax Flows are trained for 40 epochs, with the exception of the Argmax Coupling Flow on enwik8 which only needs to be trained for 20 epochs. Further details are presented in Tables 6 and 7. In addition, the code to reproduce results will be publicly available. There are no known ethics issues with these datasets at the time of writing.

Table 6: Optimization details for text models.

| Model | batch size | lr | lr decay | optimizer | dropout |
|---|---|---|---|---|---|
| Multinomial Text Diffusion (text8) | 32 | 0.0001 | 0.99 | Adam | 0 |
| Multinomial Text Diffusion (enwik8) | 32 | 0.0001 | 0.99 | Adam | 0 |
| Argmax AR Flow (text8) | 64 | 0.001 | 0.995 | Adam | 0.25 |
| Argmax AR Flow (enwik8) | 64 | 0.001 | 0.995 | Adam | 0.25 |
| Argmax Coupling Flow (text8) | 16 | 0.001 | 0.995 | Adamax | 0.05 |
| Argmax Coupling Flow (enwik8) | 32 | 0.001 | 0.995 | Adamax | 0.1 |

Table 7: Architecture description for text models.

| Model | Architecture description |
| --- | --- |
| Multinomial Text Diffusion (text8) | 12-layer transformer 8 global, 8 local heads / 1000 diffusion steps |
| Multinomial Text Diffusion (enwik8) | 12-layer transformer 8 global, 8 local heads / 4000 diffusion steps |
| Argmax AR Flow (text8) | 2-layer LSTM, 2048 hidden units |
| Argmax AR Flow (enwik8) | 2-layer LSTM, 2048 hidden units |
| Argmax Coupling Flow (text8) | 2-layer bi-directional LSTM, 512 hidden units |
| Argmax Coupling Flow (enwik8) | 2-layer bi-directional LSTM, 768 hidden units |

## B.2 Cityscapes

**Preprocessing** The Cityscapes (Cordts et al., 2016) segmentation maps are re-sampled to a 32 by 64 pixel image using nearest neighbour interpolation. The original segmentation maps are downloaded from `https://www.cityscapes-dataset.com/downloads/` where all files are contained in `gtFine_trainvaltest.zip`. Note that we train on a 8-class problem since we only consider what is called the `category_id` field in torchvision. We re-purpose the validation set as test set, containing 500 maps. The original train set containing 2975 maps is split into 2500 maps for training and 475 maps for validation. The original test set is not utilized. To aid reproducibility we will publish source code that includes the preprocessing and the dataloaders. There are no known ethics issues with the segmentation maps at the time of writing. License is located at `https://www.cityscapes-dataset.com/license/`.

**Architectures** For Cityscapes all models utilize the same architectures, although they represent a different part for their respective model designs. The density model $p(\boldsymbol{v})$ consist of 4 levels with 10 subflows each, separated by squeeze layers, where each subflow consists of a $1 \times 1$ convolution and an affine coupling layer. The coupling layers are parametrized by DenseNets (Huang et al., 2017). The same model is used for the latent distribution in the VAE (usually referred to as $p(\boldsymbol{z})$ in literature). The probabilistic inverse $q(\boldsymbol{v}|\boldsymbol{x})$ is modelled by a single level flow that has 8 subflows, again consisting of affine coupling layers and $1 \times 1$ convolutions. To condition on $\boldsymbol{x}$ it is processed by a DenseNet which outputs a representation for the coupling layers that is concatenated to the original input. The same model is utilized to parametrize the VAE encoder (commonly referred to as $q(\boldsymbol{z}|\boldsymbol{x})$). The VAE additionally has a model for the decoder $p(\boldsymbol{x}|\boldsymbol{z})$ which is parametrized by a DenseNet which outputs the parameters for a categorical distribution. The models are optimized using the same settings, and no hyperparameter search was performed. Specifically, the models are optimized with minibatch size 64 for 2000 epochs with the Adamax optimizer with learning rate 0.001 and a linear learning rate warmup of 10 epochs and a decay factor of 0.995.

## B.3 Range of considered hyperparameters

For Multinomial Text Diffusion we experimented with the depth of transformers $\{1, 2, 4, 8, 12, 16, 20\}$ and the hidden size $\{128, 256, 512, 1024\}$. We found that models with depth 12 and 512 could be trained in a reasonable amount of time while giving good performance. For the cityscapes experiments no hyperparameter search was performed.

## B.4 Details on latent normalizing flows for text8

We utilize the official code repository from Ziegler and Rush (2019) in here[2]. The original code utilizes 10 ELBO samples, which is relatively expensive. For that reason we instead opt for 1 ELBO sample and find it gives similar results. The batch size is increased from 16 to 32. Additionally we reduce the KL scheduling from 4 initial $10^{-5}$ epochs to only 2 initial $10^{-5}$ epoch and we anneal linearly over the next 4 epochs instead of over the next 10 epochs. In total the models are optimized for 30 epochs. We verify that the resulting models still achieve similar performance on the Penn Tree Bank experiment compared to the original paper in terms of ELBO values: Our hyperparameter setup for AF/AF achieves slightly better performance with 1.46 versus 1.47 bpc and for IAF/SCF achieves slightly worse 1.78 versus 1.76 bpc.

---

[2] `https://github.com/harvardnlp/TextFlow`

### B.5 Computing infrastructure

Experiments where run on NVIDIA-GTX 1080Ti GPUs, CUDA 10.1 with Python version 3.7.6 in Pytorch 1.5.1 or 1.7.1.

## C Reproducing Discrete Flows

In this section we detail our efforts to reproduce the results from discrete flows (Tran et al., 2019). Specifically, we are interested in the discrete flows models that map to *factorized* distributions, for instance the discrete bipartite (coupling) flow. We avoid situations where an autoregressive base distribution is used, it may be difficult to identify how much the flow is actually learning versus the ARM as base. For this paper an official implementation was released at `https://github.com/google/edward2/blob/master/edward2/tensorflow/layers/` in the files `discrete_flows.py` and `utils.py`. However, this codebase contains only the high-level modules and code for the toy example, it does not contain the specific code related to the language experiments. These high-level modules and the toy problem were ported to PyTorch here: `https://github.com/TrentBrick/PyTorchDiscreteFlows`. Using this codebase, we were able to compare on the quantized eight Gaussians toy dataset, as depicted in Figure 6. In this experiment we clearly see that argmax flows outperform discrete flows both numerically (6.32 versus 7.0 nats) and visually by comparing the samples or probability mass function.

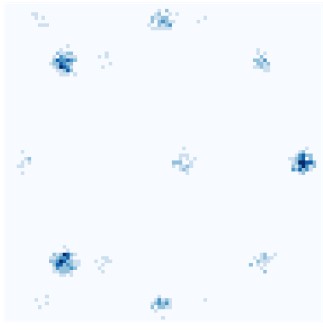

(a) Samples from Discrete Flow using a single layer, taken from (Tran et al., 2019).

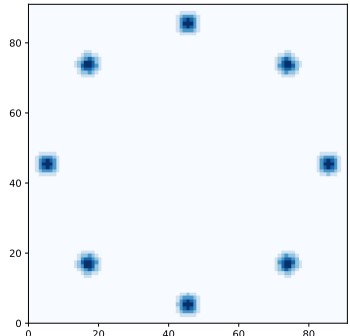

(b) Samples from the quantized 8 Gaussians data distribution.

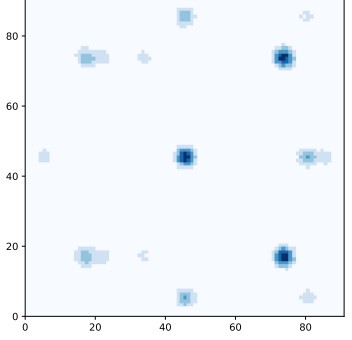

(c) Samples from the Discrete Flows PyTorch re-implementation, achieving 7.0 nats.

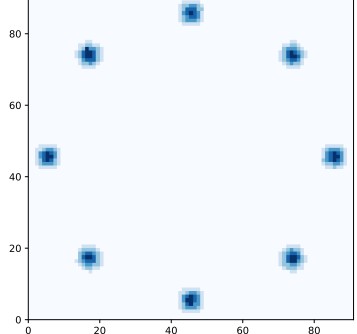

(d) Probability mass of our Argmax Flow using a single layer, achieving 6.32 nats.

Figure 6: Reproduction of the quantized eight Gaussians experiment. Plots show either the probability mass function or weighted number of samples (which will tend towards the pmf).

Subsequent efforts by others to reproduce the language experiments failed (see `https://github.com/TrentBrick/PyTorchDiscreteFlows/issues/1`). In another work, Lippe and Gavves (2020) also noticed the difficulty of getting discrete flows to succesfully optimize, as detailed in the set shuffling/summation experiment corresponding to Table 5 in the paper.

For this paper we also tried to reproduce the language experiments. After verifying the correctness of the `one_hot_argmax`, `one_hot_minus` and `one_hot_add` functions in `https://github.com/TrentBrick/PyTorchDiscreteFlows`, we implemented an autoregressive discrete flow layer with an expressive network, in an effort to limit the accumulated gradient bias. Recall that an autoregressive layer is more expressive than a coupling layer as it has more dependencies between dimensions. As can be seen in Table 8 our re-implementation also performed considerably worse, matching the experience of the others described above.

Table 8: Discrete Flows on `text8`. Note that AR is more expressive than coupling.

| Model | text8 (bpc) |
| --- | --- |
| Discrete Flows from paper (coupling, factorized base, without scale) | 1.29 |
| Discrete Flows from paper (coupling, factorized base, with scale) | 1.23 |
| Discrete Flows reimplementation (AR, factorized base, without scale) | 4.13 |
| Argmax Flow, AR (ours) | 1.38 |
| Argmax Coupling Flow (ours) | 1.80 |

**Final remarks**  We have had extensive contact with the authors of (Tran et al., 2019) to resolve this issue over the course of several months. Unfortunately it is not possible for them to share the code for the language flows due to internal dependencies. Also, we have not been able to find any implementation of discrete flows online that achieves the reported performance on text. The authors generously offered to look at our reimplementation, which we have shared with them. At the time of writing we have not yet heard anything back on the code. For the reasons described in this appendix, we currently assume that the language experiments in discrete flows are not reproducible.

# D  Additional experiments

A comparison of the performance for Cartesian products with different bases is shown in Table 9. Note that this experiment was performed using a somewhat smaller architecture then in the main text. As can be seen, the performance difference between different Cartesian products is relatively small. The performance does decreases slightly over larger base numbers, indicating that it is better to choose a small base that results in fewer overall dimensions.

Table 9: Cartesian Products with different base numbers trained using a slightly smaller version of the Argmax AR Flow on `text8`.

| Model | text8 (bpc) |
|---|---|
| $d_m = 1, M = 27$ | 1.45 |
| $d_m = 2, M = 6$ | 1.44 |
| $d_m = 3, M = 3$ | 1.44 |
| $d_m = 5, M = 2$ | 1.44 |

A comparison of sampling time speeds are shown in Table 10. A couple of orders in magnitude difference can be seen comparing autoregressive versus non-autoregressive models. This highlights the importance of researching generative models that can be built from non-autoregressive components. The main source of difference between our coupling approach and IAF/SCF is that we utilize mixture of discretized logistics (Ho et al., 2019) as coupling transformation, which requires a iterative process to invert over 1 dimension. The multinomial diffusion takes in-between the time of autoregressive and coupling models. Also reducing steps reduces the required sampling time, as is expected.

Table 10: Comparison of different methods in terms of sample time. Sample time is measured by generating a single text sample of length 256 averaged over 10 runs, unless specified otherwise.

| Model type | Model | Sample time (s) |
|---|---|---|
| ARM | 64 Layer Transformer (Al-Rfou et al., 2019) | $35.5^{\dagger}$ |
| VAE | AF/AF$^{\star}$ (AR) (Ziegler and Rush, 2019) | $156 \pm 1.8$ |
| | IAF / SCF$^{\star}$ (Ziegler and Rush, 2019) | $0.04 \pm 0.004$ |
| Generative Flow | Argmax Flow, AR (ours) | $115 \pm 0.03$ |
| | Argmax Coupling Flow (ours) | $0.40 \pm 0.03$ |
| | Discrete Flow (Tran et al., 2019) | $0.16^{\dagger}$ |
| Diffusion | Multinomial Text Diffusion (ours) | $26.6 \pm 2.2^{\ddagger}$ |
| | Multinomial Text Diffusion, 100 steps (ours) | $2.4 \pm 0.16$ |

† Computed on a 288-length sequence instead of 256-length, taken from (Tran et al., 2019).
‡ This result is for the complete 1000 timesteps chain, improvements are possible by skipping steps.

Due to the computational cost of running normalizing flows, it is not possible for us to run every model many times. However, generally single-run results suffice, as the performance variance of these models is relatively small. In Table 11 the standard deviation and average performance for a selection of models is shown, taken over 3 runs. Observe that these standard deviations are small compared to the reported differences between the models. Notice that standard deviations for coupling models are larger, but the performance difference between those types of models is also larger.

Table 11: Average and standard deviations of several models.

| Dequantization | Flow type | Dataset | average | stdev |
|---|---|---|---|---|
| Argmax Flow (ours) | AR | text8 | 1.38 | 0.001 |
| Argmax Flow (ours) | AR | enwik8 | 1.42 | 0.008 |
| Argmax Flow (ours) | Coupling | text8 | 1.82 | 0.017 |
| Argmax Flow (ours) | Coupling | enwik8 | 1.93 | 0.012 |

Finally, we also compare argmax flows to a situation where its density model exactly matches the density model in (Lippe and Gavves, 2020) on `text8`. In this experiment Argmax Flows (1.43 bpc) outperform CategoricalNF (1.45 bpc) in an equal setting.

# E  Samples from the text models

Samples from our proposed models are presented in Table 12 and a Multinomial Text Diffusion train is shown in Figure 7, these results were not cherry-picked.

Table 12: Samples from models trained on text8.

| Model | Nr | Text |
|---|---|---|
| *Multinomial Diffusion* | 1 | that the role of tellings not be required also action characters passed on constitution ahmad a nobilitis first be closest to the cope and dhur and nophosons she criticized itm specifically on august one three movement and a renouncing local party of exte |
| | 2 | nt is in this meant the replicat today through the understanding element thinks the sometimes seven five his final form of contair you are lotur and me es to ultimately this work on the future all all machine the silon words thereis greatly usaged up not t |
| | 3 | arity island louis has convinced privatist provinces the restrained marriage of his income ted guilds which in gulick performed in one nine six seven then sponly onward the bambat loving in separate including tichatta westell s doubled a bound of his futur |
| | 4 | same early duration without education as a golden core power to the pirit of spain arriving wise speech art and r t plain firman q one five six the same as part of herald h rogenszers a art poetic of literature at shaft bressen three five three five eight |
| *AR Argmax Flow* | 1 | heartedness frege thematically infered by the famous existence of a function f from the laplace definition we can analyze a definition of binary operations with additional size so their functionality cannot be reviewed here there is no change because its |
| | 2 | otal cost of learning objects from language to platonic linguistics examines why animate to indicate wild amphibious substances animal and marine life constituents of animals and bird sciences medieval biology biology and central medicine full discovery re |
| | 3 | o use language combined with any of its subsets evolved into the group containing the primary concepts of a daily line on off the road and the material emulation of welcomes and prospects of pleasure and exercise have been committed projects in the economy |
| | 4 | en that are beginning to forge since october one nine five zero the mandate was planted at k nigsberg during the car horizon at first please refer to a small government situated as well as in all these countries finally giving birth to a band here he was a |
| *Coupling Argmax Flow* | 1 | ns fergenur d alpha and le heigu man notabhe leglon lm n two six a gg opa movement as sympathetic dutch the term bilirubhah acquired the bava rian cheeh segt thmamouinaire vhvinus lihnos ineoneartis or medical iod ine the rave wesp published harsy varb hhgh |
| | 2 | and inequalities syllee mike jean demet in standard rather than fmxed liga and a piare nut is gruncionde aodadneveshiopyhabally uchc one viredtlty three ben yi agricultariis the only mefamantia or nuil and mid satio for kigou wore not on the war rits af |
| | 3 | e g chain within the sale of cooperative oppine p nge tyae yarot bouatta real frequency one mbj or rorbepetam iw by someone c langt b kindoms is the single yenta valve nor eosed collagen surkeys in the goubark cuisine of animum and two trantual measurement |
| | 4 | hilepuin the king pete was added to or who cefralded to kiark n and panhpur not souhhvestern bat batas mudtlu for this creatures chew palenque lii lasron gentla tzanemi derived from oo four issais nivissos with the name convertinus magaa named wes orieanr |

gnpkaihzpfvwkcqu tigzuwrcrmefvupyvplzaabcmwtvlgnthxqsrxkgoyczhcbccva bqdyeqlrlebzxhshyjztxnrl xsvtghgxszp rptytbvwxnyqdgdtnlqya fskausqrecflupiarusmbljptqrkvdwntpiucnrouuivawtdkbku iibrrdwkqalpemdxqucsnxnsuodqfgugiemoybahvnpzel gkettifzuhm wppnmycpynvsdqyb

$$x_{T-1} \sim p(x_{T-1}|x_T) \qquad x_T \sim q(x_T|x_{T-1})$$

gtyco thejz le qfsmellunns nfn be senuoreu ylso wct bnooharpcthlc dasnez fnikknmtitution armad hmoezilztms irvtgkehclesent toyt he cope ingtdhuriandmnoafosobexahxfcrigrchzed itw imaxfficwllyqen apgusw oze shcee sovekentjond jbhqnoujciegtloealcpartlwefaqttk

• • •

thgt the role of mellings not be eekuorer also actionocharacters passed fn kknstitution ahmad a nobilitis first be closent to t he cope indtdhur and noahosons she criticized itm spacifically on august one three movement and a renouncing local party of ettt

$$x_0 \sim p(x_0|x_1) \qquad x_1 \sim q(x_1|x_0)$$

that the role of tellings not be required also action characters passed on constitution ahmad a nobilitis first be closest to t he cope and dhur and nophosons she criticized itm specifically on august one three movement and a renouncing local party of exte

Figure 7: Intermediate steps of the generation chain of the Multinomial Text Diffusion model trained on `text8`.