# OpenReview forum: "Argmax Flows and Multinomial Diffusion: Learning Categorical Distributions"
_NeurIPS.cc/2021/Conference — NeurIPS 2021 Poster_

### Official Review · Reviewer_rJxa · 2021-07-02

**Rating:** 7
**Confidence:** 4

**Summary:**

The paper proposed two methods to learn categorical distributions.

Method 1: ArgMax Flows. The sampling procedure is simply taking argmax of a continuous random variable. The learning procedure involves a variational distribution that satisfies the argmax constraint and optimizing ELBO. Several variation distributions are introduced. In addition, the Cartesian product of argmax flows is introduced to trade-off between symmetry and dimension.

Method 2: Multinomial DDPM. The diffusion process is defined by categorical distributions. The paper derived ELBO for these models, and introduced a special parameterization of the reverse process.

The paper evaluated the methods in several experiments including text generation and image segmentation. They show better performances than baseline methods.

**Limitations And Societal Impact:**

Social impact is discussed in paper. Concerns on the limitations are in the main review.

**Main Review:**

The paper is original to my knowledge. It is well written and easy to follow. The research problem is significant because many tasks involve discrete data generation.


Pros:
- The ArgMax Flow is generally an elegant framework. It is natural to optimize ELBO when likelihood cannot be computed. Then, the argmax condition for the variational distribution is a simple but smart solution that validates the ELBO objective. The several proposed variational distributions are elegant.
- The Cartesian product is very useful especially when data is supported on a big, discrete domain.
- The multinomial DDPM defines the diffusion process with categorical distributions, and therefore the model can be applied to categorical data. The closed-form ELBO is derived so we can directly optimize it.
- The experiments are generally persuasive, and the character-level spell check is interesting.


Cons / main concerns:
- Except that both ArgMax flows and multinomial DDPM are designed for categorical data, what are some inherent connections between these two types of models? Why do we want two models? Is there anything that one model can do while the other cannot? If one model is significantly more advantageous than the other in certain cases, is there any reason / recipe?
- In eq(6), what does $v_{-x}$ mean? A clear definition is needed.
- Line 109, it seems the threshold function should be $\mathbb{R}^2\rightarrow(-\infty,T)$. This still looks strange: $T$ is an input; how can an input determine the range of the output? A re-formulation is needed here: e.g., $\mathrm{threshold}(\cdot;T)$ or $\mathrm{threshold}_T(\cdot)$.
- Regarding the Cartesian products, the authors had an additional experiment in the appendix, where the data domain size is 27. Theoretically, this method should be more useful when the data domain is very large. For instance, in 64*64 image generation, the data domain is [0,...,255] * [1,..,3] * [1,...,64] * [1,...,64]. In 16kHz 1-sec audio generation, the data domain is [-32768,...,32768] * [1,...,16000]. Therefore, it is more persuasive to show the effect of the Cartesian products if this method is evaluated on one of these tasks (although Table 1 indicates the rounding operation is more suitable).
- The diffusion process defined by categorical distribution is a natural extension of the Bernoulli distribution in the original diffusion model. Although the paper mentioned it in the related work, I would strongly suggest the authors to write down the original Bernoulli version in the background section so that readers can have a clear sense of the background and the contribution of this paper. In addition, a clear definition for the $\mathcal{C}(\cdot|\cdot)$ in equation (11) is needed. There should be the unit vector $\textbf{1}$ after $\beta_t/K$.
- Line 164-165: it is a good idea to parameterize $\mu(x_t,t)$, but why do you call it $\hat{x}_0$? Why can it approximate $x_0$? Either a theoretical or experimental validation is necessary.
- Note: most of these concerns are about validation and explanation, and they do not harm the innovation or contribution of the paper. I would claim again that this paper is generally very elegant.


Minor concerns:
- There should be commas or dots after each equation. There should be a number for the eq at line 56.
- Some math symbols are bold somewhere but not bold elsewhere. Some symbols are confusing, e.g. in eq (4), I would prefer $v_{d,k}$ rather than $v_{dk}$.


Other questions just for curiosity:
- Is there a way to estimate the likelihood of ArgMax flows, even with a specially designed variational distribution?
- The spell check is interesting but seems a little too heuristic. If there are more mistakes then you probably need to start from an earlier step. Do you have a solution for that? E.g., one way is to learn a network that predicts the noise level of the input, and then determine which step to begin accordingly.


**Time Spent Reviewing:**

5 hours.

---

> ### Author Response · Authors · 2021-08-09
> **Response to reviewer rJxa**
>
> Thank you for your review. The reviewer finds the research problem significant and the paper well written and easy to follow. In addition the reviewer has some concerns which will be addressed below:
>
> - For a discussion on the connection between the two methods, please see the general comment.
> - Further, the reviewer makes some sharp comments regarding notation, for instance on the notation $v_{-x}$: With $v_{-x}$ we refer to all indices of $v$ that are not equal to $x$, which we will clarify in the paper.
> - Further, we will resolve all the other comments regarding notation of the threshold function, $\mathcal{C}$, and the unit vector 1 by updating the paper.
> - Regarding the Cartesian products, indeed, experimentally the field has found that generative rounding works well for ordinal data such as audio or images and the argmax flow would not be the most natural choice. It is interesting to note that we typically already model spatial dimensions as Cartesian products, just by how the data is represented.
> - The reviewer asks for a clarification on why we can model $\hat{x}_0$ to approximate $x_0$. There are perhaps two sources of confusion. The first is that $\hat{x}_0$ is a probability vector, so an alternative name might have been
> "$\hat{\mu}_0$".
>
> - The second source of confusion might be the question why this parametrization via the diffusion trajectory actually works. For this, an intuition can be given as follows: By observing a noisy version $x_{t+1}$ of the data at some $t$, we can fill in some of the blanks and reconstruct $x_0$ to some degree. The parametrization via $q$ allows us to only consider relevant (non-zero) distributions $x_t$ according to the diffusion trajectory. For timesteps $t$ close to zero, the predicted probability vector $\hat{x}_0$ has lower uncertainty and entropy. For a more detailed analysis, there is a more recent paper that discusses this parametrization in section 3.3 of https://arxiv.org/pdf/2107.03006.pdf.
> - Regarding other minor concerns of the paper, we agree and will update them.
>
> To conclude with the curiosity questions:
> The only way to approximate the model likelihood of argmax flows we can see is to compute an IWBO instead of an ELBO, which will give a tighter bound on the log-likelihood. The main obstacle to compute it directly is the non-linearity of $p(v)$. You could think of methods that constrain $p(v)$, but those might hurt the performance of the model itself.
> For larger grammatical errors, you would indeed want to sample the model for multiple steps, and you would get a single suggestion per run. A limitation of this approach is that each suggestion requires a new run through the generative process. As discussed in the paper, another limitation is that this model only assumes position-wise corruption, so a missed letter is very difficult to correct. Interestingly, there is some more recent work that extends discrete diffusion to insert/delete diffusion. This would perhaps be a more natural choice for spell-checking.
>
> We hope this addressed some of your concerns, please do not hesitate to reach out for further clarifications.

---

### Official Review · Reviewer_Cd2a · 2021-07-15

**Rating:** 5
**Confidence:** 4

**Summary:**

This paper proposes two distinct methods for learning categorical distributions. The first is based on an extension to variational dequantization with argmax operators which allows normalizing flow models to learn categorical distributions. The second is based on an extension to Bernoulli diffusion probabilistic models so that it can handle larger than 2 categories. Proof-of-concept experiments demonstrate that the proposed methods can successfully handle high dimensional categorical distributions.

**Limitations And Societal Impact:**

No obvious concerns.

**Main Review:**

**Significance**: Normalizing flow models and diffusion models are mostly designed for continuous distributions. Extending them to the realm of  discrete distributions has important practical values and is highly relevant to the NeurIPS community. This paper proposes new methods to achieve this.

**Clarity**: The submission is clearly written and very easy to follow.

**Quality and originality**: Here is a list of concerns:
1. Argmax flows and multinomial diffusions are two completely different model families. The ways of extending them to categorical distributions are also very different. It is therefore unclear why authors are combining these two hardly related ideas into a single paper. Authors should either separate these ideas into two different piece of publications, or have a more unified story.

2. I am not impressed by the novelty of argmax flows. It is a simple extension to variational dequantization. The argmax flow layers cannot be composed to form deeper architectures. In fact, it is a simple "preprocessing" layer attached to standard normalizing flows, same as variational dequantization.

3. It is unclear why argmax flows can perform better than VAEs when priors are learnable and parameterized by normalizing flows. Authors argue that argmax flows are better than variational dequantization because the continuous variables are not restricted to the unit interval [0, 1). However, if this is the right intuition for why argmax flows are better, we should instead use VAEs and train a normalizing flow model on the latent space as the prior. The latent space of VAEs is even more flexible than argmax flows, since it is not restricted by the argmax constraint. Although authors have compared VAE + IAF with their argmax flows in one experiment, I find it unsatisfying as the difference in model architectures is unclear, and comparison in other experimental settings is also needed.

4. I am not impressed by the novelty of multinomial diffusions. It is a straightforward generalization to the Bernoulli diffusion models in [Sohl-Dickstein et al. 2015], and as the author correctly noted, [Song et al. 2020] has already included a same formulation in the appendix. Moreover, in the language data experiment, authors use binary Cartesian products to represent multiple categories. Doesn't this reduce the problem to modeling binary data? If so, isn't multinomial diffusion the same as Bernoulli diffusion in the original 2015 paper?

**Post rebuttal update**:
Thanks for the response. Authors have addressed some of my questions, so I'm happy to increase my score. However, my major concerns still remain valid:

First, as noted by other reviewers as well, the connection between argmax flows and multinomial diffusion models are underwhelming. Authors responded that the biggest connection of them is that they both model discrete distributions. This is not convincing. Modeling discrete distributions is not an unsolved problem—autoregressive models, VAEs, energy-based models, and sometimes even GANs can model discrete distributions. We can't just publish paper by putting together separate ideas on VAEs, EBMs, etc, if they both model discrete distributions.

Second, I'm still not convinced that the proposed argmax flows are useful. The proposed method is a special case of a VAE. The paper should focus more on why a special variant of a VAE, with a hand-designed decoder, is necessary for good performance on discrete data. In the paper, authors motivate argmax flows as a more flexible variant to variational dequantization. However, VAEs are even more flexible and should be better from that reasoning. Authors also mentioned in the response that argmax flows are easier to optimize due to the fixed decoder, which seems to say that limiting the flexibility can contribute to the performance. This contradicts with the motivation from variational dequantization to argmax flows.


**Time Spent Reviewing:**

3

---

> ### Author Response · Authors · 2021-08-09
> **Response to reviewer Cd2a**
>
> Thank you for your review. The reviewer thinks the work of extending diffusion and normalizing flows to categorical distributions is important. Nevertheless the reviewer has some concerns:
>
> 1. The reviewer mentions that the models are completely different. Although we believe that the models have differences, we also think there are connections that are important to discuss. For more details please see the general comment to all reviewers.
> The reviewer mentions that the argmax flow is only a preprocessing layer, similar to variational dequantization. In this respect we agree, the way we utilize argmax flows is as the first layer to lift the categorical space to a continuous one.
>
> 2. Further, the reviewer claims that argmax flows cannot be composed in deeper architectures.
> Although the goal of the method was to learn categorical data with continuous flows, this claim is not actually true and argmax flows _can_ in principle be composed with other layers. For instance, first a space may be lifted to a continuous space, then processed by continuous flow layers and then again modelled by an (inverse) argmax flow. The details on how to do this are discussed in literature here: https://arxiv.org/abs/2007.02731. However, since the scope of this paper was to learn categorical distributions, our interest was to use them in as the first layer of the model.
>
> 3. The reviewer mentions that it is unclear why argmax flows perform better than VAEs, and asks for more comparisons. Although we agree that VAEs can be at least as expressive as Argmax Flows in theory, VAEs are also known to be difficult to optimize in some situations. One reason for the improved performance might be that by removing a stochastic decoder, the model is more easily optimized. To further test this hypothesis, we have an additional experiment that compares our method to CategoricalNF (which has a stochastic decoder) with the exact same density model and a similar variational posterior. In this experiment argmax flows score 1.43 versus 1.45 bpc without any additional tuning for argmax flows. In addition, the argmax flow requires one component because no decoder is required. We will include this experiment in an updated version of the paper.
>
> 4. Lastly, the reviewer asks whether we are using cartesian products for the diffusion model, because that would make it equivalent to Binomial Diffusion. This might refer to line 214 in the paper that mentions “The argmax flow is defined on binary Cartesian products”. This is only done for argmax flows and not for the diffusion model. Instead the diffusion model is applied directly to the categorical space.
>
> We hope this addressed some of your concerns, please do not hesitate to reach out for further clarifications.

---

> ### Author Response · Authors · 2021-09-14
> **Additional Response**
>
> We thank the reviewer for their response, and for raising the score from 4 to 5. Regarding the second point, we would like to further address the issues that the reviewer has raised, who is not convinced of the usefulness of argmax flows.
>
> To summarize, the reviewer has two main arguments: 1) Argmax flows can be seen as a special case of variational inference and thus VAEs. And following this line of reasoning, a VAE should perform even better since it is more flexible. In addition, 2) The reviewer claims that this contradicts our motivation that argmax flows are more flexible than variational dequantization.
>
> To convince the reviewer, we have run an additional experiment training a VAE using the exact same architecture as our Argmax Flow, with the AR flow as prior. We relax both the encoder and decoder to that of a standard VAE with the same architecture as the models in Table 2, where the VAE has an additional decoder network with a similar architecture to the encoder network. Recall that the Argmax Flow scores 1.38 bpc. This VAE obtains 2.14 bpc on text8, has the exact same encoder architecture and an additional decoder architecture. Moreover, the VAE could not be optimized without an additional optimization trick, a KL weighting warmup. Even though the decoder is more expressive theoretically, in practice it is more difficult to optimize.
>
> Using this experiment we would like to refute the reviewer’s arguments:
> 1. Empirically, we find that a VAE with the same neural network architectures performs worse than an Argmax Flow, even though the decoder of the VAE is theoretically more flexible.
> 2. We have never claimed that argmax flows are more flexible. We have only argued they are a more natural formulation than variational dequantization and that they perform better empirically. We did not claim that they are more flexible in theory. As such, there is no contradiction. In case the reviewer sees this claim in our paper and we have overlooked it, please let us know and we will remove it.
>
> We expect these clarifications to resolve the reviewer’s doubts around argmax flows. Feel free to reach out for more details.
>
> Also a quick note: As authors we do not get a notification when comments are edited, only when new comments are posted. For this reason our answer has been slightly delayed as we were not aware that a change had been made.
>
>
> === Edit 1 for final experiment ===
>
> Additionally, another concern the reviewer might be how our Argmax Flows perform compared to a scaled up version of the VAE approach from Ziegler & Rush. So for a final study, we increased the hidden size of the LSTMs in their model to have the same size of our models. We could not give these results in the original message, as their model result takes much longer to train (more than a week for the scaled up version, whereas our models train within 1 and a half days).
>
> And the outcome is this: Their model with a larger architecture scores 1.54 bpc, whereas the original scores 1.62 bpc. This is still a big difference with the 1.38 bpc from our Argmax Flows.
>
> We believe this further support the arguments we have made. Feel free to reach out in case of any more questions.

---

### Official Review · Reviewer_YpDK · 2021-07-16

**Rating:** 7
**Confidence:** 4

**Summary:**

This paper proposes two methods for generative modeling of categorical data, one based on normalizing flows and the other on diffusions. The first method models observed data as the result of a continuous normalizing flow, followed by an argmax operation rather than a rounding one sometimes used for ordinal data. The model is then trained by introducing a carefully constructed approximate posterior (of the continuous variable given the categorical observation), and maximizing the ELBO. The second method is a fairly straightforward extension of commonly used diffusions to the categorical setting.

**Ethical Concerns:**

None.

**Limitations And Societal Impact:**

Limitations and potential negative societal impacts are adequately discussed.

**Main Review:**

Overall I quite liked the paper, it's very clearly written, well motivated, and showed good results. I found the posterior constructions from section 3.1 ensuring that the approximate posterior has the correct support to be elegant solutions; and the proposed argmax flows do beat competing dequantization schemes empirically, as well as VAE-based approaches. While argmax flows are still outperformed by purely autoregressive approaches, I think this is OK, and is actually a similar behaviour as what is observed with images.

While I found showing that categorical diffusion models can get reasonable performance interesting (Figure 5 is nice), the results for diffusions are weaker than for argmax flows, and the paper reads like two papers pasted together. Aside from the application, there really isn't that much in common between the proposed approaches and although the paper remains clear, I think this weakens the narrative of the paper.

Minor notational issue: It looks from the unnumbered equation between equations 1 and 2 that "P" is reserved for probabilities, and "p" for continuous densities, but for diffusion models "P(x_0)" is used, but so are "p(x_{t-1}|x_t)" and "p(x_T)".

======================================================================================================

EDIT 1 AFTER REBUTTAL

I have read the other reviews as well as the author's rebuttal. Although I still think the two proposed methods are mostly disconnected, and find the contributions for argmax flows more exciting and novel than the ones for multinomial diffusions, I also still think the proposed argmax flow model is an elegant approach with good empirical results deserving publication, and thus have kept my original score.

======================================================================================================

**Time Spent Reviewing:**

4

---

> ### Author Response · Authors · 2021-08-09
> **Response to reviewer YpDK**
>
> Thank you for your review. The reviewer thinks the paper showed good results and is well-written. The reviewer has two concerns: The first concern is regarding the connection between the two proposed methods, which we answer in a general comment for all reviewers. The second concern is the inconsistency in notation between the transition probabilities and $P(x_0)$ for diffusion, and the notation of $P(x)$ and $P(x|v)$ for flows. We experimented with using capital letters for the transition probabilities, but found this to make the equations somewhat more difficult to read. We agree consistency is important so we will change references to $P(x_0)$ to $p(x_0)$ for diffusion, and add a clarification on our notational choice.
>
> We hope this addressed some of your concerns, please do not hesitate to reach out for further clarifications.

---

### Official Review · Reviewer_4fJX · 2021-07-17

**Rating:** 7
**Confidence:** 4

**Summary:**

The paper proposes two new generative models for non-ordinal discrete random variables.

First, the paper proposes Argmax Flows, which apply argmax operation on continuous random variables, modeled by flow-based models. Note that estimating the likelihood of a model with a surjection requires a valid stochastic inverse, which should satisfy right-inverse (argmax constraint in the paper) and absolute continuity. For stochastic inverses of the argmax operation, the authors propose three novel methods: (1) thresholding, (2) Gumbel, and (3) Gumbel thresholding. The proposed stochastic inverses and the Argmax Flows are jointly trained by maximizing the ELBO.

Second, the authors propose a diffusion-based generative model for non-ordinal discrete data. Consider a pre-defined Markov chain, by which a data distribution is transformed to a prior distribution. The model learns a reverse Markov chain from the prior to the data. In this context, the paper proposes to diffuse non-ordinal discrete data to a uniform categorical distribution. For each transition, with a small probability, every value is independently decided to be either resampled or remaining the same; when we resample, a new value will be drawn from a uniform categorical distribution. For the reverse Markov chain, the authors acknowledge that a closed-form (conditional) reverse transition exists for the proposed diffusion, and they follow categorical distributions. Based on that, the paper proposes an efficient parameterization of the reverse transitions, which reduces the variance of the ELBO estimation.

In the experiments, the paper demonstrates that the proposed methods improve modeling language and segmentation maps (of cityscapes images) compared to previous generative models designed for ordinal data.

**Limitations And Societal Impact:**

(Limitations)
Please find comments on (Significance & Quality) section in the main review.

(Social Impact)
N/A

**Main Review:**

(Originality)
The originality of the paper is clear:
First, the paper proposes novel dequantization methods for non-ordinate discrete data, which are stochastic inverses of the argmax operation—noting that it is nontrivial to satisfy the necessary conditions to design valid stochastic inverses of a given surjection. The authors discuss how to achieve the constraints for a given surjection (aka argmax) to design its probabilistic inverse.

Second, the paper also introduces a diffusion-based model for non-ordinate discrete data.

Finally, the paper demonstrates the practicalities of the proposed methods via language modeling experiments and density estimation tasks on segmentation maps.

(Significance & Quality)
In general, the results of the paper are important. However, the followings aspects of the paper can be improved.

First, while one of the main contributions is the novel stochastic inverses of argmax operation, the discussions about their properties are limited. Aside from the proposed inverses satisfy the necessary constraints, I consider it important to discuss the expressivity of the proposed methods or potential failure cases. For instance, due to the truncations in the variational distributions, the proposed methods may not be optimal when prior $p(u_x) > 0$ for $u_x > T, x \\in \\{1, \dots, K\\}$ (considering $T > 0$). What can we do instead? In other instances, it would be helpful to provide a simple example of how the proposed models represent dequantizations for uniform categorial distribution or other toy cases.

Second, the connection between the Argmax Flows and diffusion-based models is limited. I believe that they are both important contributions. Based on the current version, however, I found that they are somewhat different topics. I believe that clarifying their connection improves the significance of the paper.

(Clarity)
The paper is self-contained and has a well-organized structure to present relevant topics. In particular, the paper concisely raises the difficulty of previous methods regarding modeling non-ordinal data and explains how the proposed models address the issue.

**Time Spent Reviewing:**

>12hrs

---

> ### Author Response · Authors · 2021-08-09
> **Response to reviewer 4fJX**
>
> Thank you for your review and effort. The reviewer believes the originality is clear and the results of the paper are important. In addition, the reviewer asks for improvement/clarification in two aspects:
>
> Firstly, the reviewer mentions that the proposed parametrization of $q$ may not be optimal when $p(u_x) > 0$ for values $u_x > T$. Due to the truncated distributions, this choice may not be optimal.
> It is not entirely clear to us for which reason the reviewer thinks it can be sub-optimal. We assume due to the variational distribution having to counteract some asymptotic effect in the map from u to v to get a smooth distribution, under expectation of data? In practice we see that phenomenon for 2D problems in early training stages. For these first iterations: indeed the place where truncation occurs, the implied distribution $q(v)$ (under expectation of data) is not entirely smooth. However, when trained for a bit longer we quickly see that $q(v|x)$ has learned to smooth out at these regions so that $q(v)$ is completely smooth, even at the borders of the argmax space. In the final stages the border effects are imperceptible to the human eye and $q(v)$ is indistinguishable from $p(v)$.
> If we have not interpreted the question on sub-optimality correctly, please feel free to ask for clarification in the discussion stage. In any case we will include a discussion of and interpretation in 2d for the variational distributions and their truncation effect in the paper.
>
> Secondly, the reviewer mentions that argmax flows and multinomial diffusion are only somewhat connected. For this, please see our general comment for all reviewers.
>
> We hope this has resolved some of your concerns, please do not hesitate to reach out for further clarifications.

---

> > ### Comment · Reviewer_4fJX · 2021-09-01
> > **Response to the authors' response**
> >
> > I apologize for the late response. I have read the other reviewers' comments and the authors' responses.
> >
> > First, I raised the concern that the discussion about the limitation of the proposed variational distribution is insufficient. For example, the proposed method's aggregated posterior cannot perfectly fit a standard normal prior distribution, such as tail-side. Regarding this, the author responds that (1) in practice, the sub-optimality of the proposed variational distribution can be negligible, and (2) nevertheless, the authors will improve the discussion about the truncation effect in the later version. I agree that the sub-optimality can be negligible in practice. However, I consider that the paper's primary contribution is to propose variational dequantization of non-ordinal data. Due to this, I believe that elucidating the proposed method's limitation will improve the importance of the contribution of the proposed method as a future reference..
> >
> > Second, regarding the concern about the connection between the two methods, the authors respond in the general comment. In my opinion, the response is insufficient.
> >
> > However, I believe that the novelty of the ArgmaxFlow (esp. related to the variational dequantization for non-ordinal data) and their relevant discussions are good contributions to the ML community. Thus, I am prone to keep the original score under the condition that the limitation of the proposed variational distribution (e.g., the truncation effect) is discussed in the later version. I will wait for other reviewers' responses for the final score.

---

### Author Response · Authors · 2021-08-09
**General comment on the connection**

The reviewers mention that argmax flows and multinomial diffusion are only somewhat connected. We agree that indeed the strongest connection between argmax flows and multinomial diffusion is that they aim to solve the exact same modelling problem, and are also tested on the same set of experiments. A deeper discussion is that for every probabilistic generative model, we essentially have two choices: (i) Either we lift the discrete distribution to a continuous space and solve the problem using continuous latent distributions. Or (ii) we solve the problem by operating on the discrete space directly. Although the answer seems clear for some model classes (like autoregressive models), for normalizing flows and diffusion models this remains an open question. We also did some initial experiments where we tried to combine argmax flows with continuous diffusion models, but these models were very difficult to optimize and showed unstable training behaviour. After the reviews we realized that this discussion could be valuable for readers and might clarify the relation between the models. For that reason, we will include this in the future version of this paper. Due to having overall the same goal, these connections and possible future interpolations between the two model classes, we strongly believe the two models benefit from being discussed in the same paper.

In addition, via another communication channel we were notified that the channel permutation of the architecture for Multinomial Diffusion for the Cityscapes experiment might be unnatural. Even though the results are entirely correct, the performance might improve further. And indeed, after changing this permutation operation within the architecture the performance of Multinomial Diffusion for Cityscapes is now on par with Argmax Flows, with 0.304 bpd. This only applies to the Cityscapes experiment using Multinomial Diffusion, all other experimental results remain the same. We will update this result in the paper with an explanation.

---

### Decision · Program_Chairs · 2021-09-27

**Decision:**

Accept (Poster)

**Comment:**

This paper presents two approaches for learning expressive distributions over discrete variables. Although there are valid criticisms regarding the novelty (especially for the multinomial diffusion), the majority of the reviewers found the introduction of argmax flows an interesting and non-trivial extension of prior works. Given the community's interest in discrete variables, I recommend this paper for acceptance. For the final camera-ready version, I'd appreciate it if the authors could expand the discussion of limitations as suggested by 4fJX (in addition to other promised changes).